# Efficient, Property-Aligned Fan-Out Retrieval via RL-Compiled Diffusion

Pengcheng Jiang [1 2 *]   Judith Yue Li [1]   Moonkyung Ryu [1]   R. Lily Hu [1]   Kun Su [1]   Zhong Yi Wan [1]   Liam Hebert [1]
Hao Peng [1]   Jiawei Han [2]   Dima Kuzmin [1]   Craig Boutilier [1]

## Abstract

Many modern retrieval problems are *set-valued*: given a broad intent, the system must return a *collection* of results that optimizes higher-order properties (e.g., diversity, coverage, complementarity, coherence) while staying grounded to a fixed database. Set-valued objectives are inherently non-decomposable and are not captured by existing supervised (query, content) datasets which only prioritize top-1 retrieval. While reinforcement learning (RL) can optimize set-level objectives via interaction, deploying an RL-tuned LLM for fan-out retrieval is prohibitively expensive at query time. Conversely, diffusion-based generative retrieval enables efficient single-pass fan-out in embedding space, but requires objective-aligned training targets. To address these issues, we propose R4T (Retrieve-for-Train), which uses RL *once* as an objective transducer in a three step process: (i) train a fan-out LLM with composite set-level rewards, (ii) synthesize objective-consistent training pairs, and (iii) train a lightweight diffusion retriever to model the conditional distribution of set-valued outputs. Across Polyvore and a music playlist dataset, R4T improves retrieval quality over strong baselines while reducing query-time fan-out latency by an order of magnitude.

## 1. Introduction

Retrieval systems are increasingly expected to return *sets* of results rather than a single best match. In many real-world applications, the desired output is a collection that jointly satisfies higher-order properties: a search interface may expand a broad query into multiple intents to improve coverage; a recommender may produce a slate that is diverse yet coherent; and a bundling system may retrieve complementary items that collectively meet the demands of complex queries. These requirements have motivated *generative retrieval* formulations, where candidates are *produced* by a model rather than simply selected (e.g., using only nearest-neighbor similarity) (Tay et al., 2022; Tomasi et al., 2025; Deffayet et al., 2023).

A central challenge is that many of these problems are set-valued and non-decomposable. Unlike classical retrieval tasks with a single, labelable correct item, fan-out retrieval often admits no unique ground truth: many different item sets can be valid for the same broad intent. As a result, retrieval quality is defined by *set-level* properties (e.g., diversity, intent coverage, complementarity, and stylistic coherence), while still requiring strict *groundedness* to the target database. This makes the standard supervised paradigm more challenging in practice: collecting property-aligned (query, content) pairs that explicitly encode these set-level objectives is costly, subjective, and frequently infeasible, especially for domain-specific or customized corpora.

*Reinforcement learning (RL)* is a natural framework for optimizing such behaviors through reward-driven interaction with a database (Zhang et al., 2025a; Jiang et al., 2025b), and recent work shows RL can shape retrieval policies beyond pointwise relevance (Jiang et al., 2025a;c). However, directly deploying RL-trained language models at inference time is often impractical: autoregressive fan-out generation with repeated retrieval calls incurs substantial latency, and set-level rewards can exhibit high-variance and are prone to shortcut exploitation. These issues complicate stable deployment and motivate separating reward-driven optimization from inference-time retrieval.

In parallel, *diffusion-based generative retrieval* enables efficient, non-autoregressive sampling in embedding space (Bao et al., 2025; Guinot et al., 2025; Tomasi et al., 2025). This shifts retrieval from heavy, "System 2" autoregressive generation to parallel "System 1" sampling, allowing entire result "slates" or "bundles" to be generated in a single pass. Yet diffusion retrievers critically depend on large amounts of *property-aligned training targets*, precisely the resource that is scarce or ambiguous in non-decomposable, set-valued retrieval tasks.

---

*Work done as a Student Researcher at Google Research. [1]Google Research [2]University of Illinois Urbana Champaign.

*Proceedings of the 43rd International Conference on Machine Learning*, Seoul, South Korea. PMLR 306, 2026. Copyright 2026 by the author(s).

This paper addresses the resulting bottleneck: *retrieval objectives have become richer than the supervision available to train efficient fan-out retrievers under those objectives.* Our key idea is to use RL not as the deployed inference mechanism, but as a one-time *objective transducer* that converts complex set-level reward specifications into scalable supervision.

We propose **R4T (Retrieve for Train)**. R4T operates in three stages: (1) we train a fan-out policy implemented as a language model using RL with composite rewards that explicitly encode desired set-level properties; (2) we use the optimized policy to synthesize objective-consistent training data by harvesting and filtering successful trajectories; and (3) we train a lightweight, single-pass generative retriever on this synthesized data. In our instantiation, this retriever is a coherent embedding diffusion model that performs efficient, controllable fan-out retrieval at inference time.

**Contributions.** We make three primary contributions. First, we present a general framework for compiling reward-optimized behaviors over set-valued, non-decomposable retrieval objectives into data for supervised training. Second, we instantiate this framework using Soft-GRPO for fan-out policy optimization and coherent embedding-based diffusion for single-pass generation at inference time. Third, we demonstrate the effectiveness of R4T in two distinct regimes: *open-ended abstract retrieval (OAR)*, where no ground truth exists and quality is defined by reward-specified set-level properties over collection-level outputs; and *weakly supervised compositional retrieval (WSCR)*, where each query admits many valid item sets and supervision is provided via weak reference sets. Across both settings, R4T improves retrieval quality over strong fan-out baselines while maintaining the efficient inference needed for production-level deployment.

## 2. Method: R4T

We introduce **R4T** (Retrieve-for-Train), a framework for training *set-valued* generative retrievers under *non-decomposable* objectives. The central idea is to use reinforcement learning (RL) *once* as an *objective transducer*; rather than deploying an RL-tuned language model at inference time, we use RL to discover reward-aligned fan-out behaviors and convert them into scalable supervision for a lightweight, single-pass retriever.

### 2.1. Problem Setup: Set-Valued Fan-out Retrieval

We work in a setting where, given a broad query $q$, our retrieval system returns a *set* of results from a fixed database $\mathcal{D}$. To handle the set-based nature of retrieval, we consider a *fan-out formulation*, where a policy first generates $k$ sub-queries $Q = \{q_1, \ldots, q_k\}$, and a fixed retriever $R(\cdot)$

executes each sub-query to retrieve candidate results. Let $\mathcal{C}_i = R(q_i, \mathcal{D})$ denote the retrieved candidates for sub-query $q_i$, and $\mathcal{R}(Q) = \bigcup_{i=1}^{k} \mathcal{C}_i$ the union of the fan-out results. Crucially, we assume that result quality is defined by *set-level* properties, such as diversity, coverage, or complementarity, which are generally non-decomposable and, moreover, admit multiple (non-unique) correct results. The latter property poses challenges for ground-truth supervision.

### 2.2. R4T Overview

R4T consists of three principal stages: The first is *RL policy optimization*, where we train a *fan-out language model (FOLM)* $\pi_\theta$ to generate sub-queries whose database interactions maximize some task-specific *set-level* reward (we discuss various rewards below). The second phase is *synthetic supervision*, in which we run the optimized FOLM policy to collect high-reward trajectories and convert them into a synthetic dataset of *set-valued* targets. In the final stage, compiled deployment trains a lightweight generative retriever to model the conditional distribution of set-valued targets given the query, enabling efficient single-pass fan-out at inference time. We detail each of these stages below (see Figure 1 for a graphical depiction).

**Two deployment variants: R4T-FOLM vs. R4T-Diffusion.** R4T separates *reward-driven discovery of fan-out behaviors* from *efficient inference-time fan-out*. We therefore evaluate two deployment variants that share the same RL optimization stage but differ in how fan-out is executed at test time.

**R4T-FOLM (RL-tuned fan-out language model).** We directly deploy the RL-optimized FOLM $\pi_{\theta^*}$ at inference time to generate $k$ sub-queries $Q = \{q_1, \ldots, q_k\} \sim \pi_{\theta^*}(\cdot \mid q)$, whose retrieved results are aggregated into $\mathcal{R}(Q) = \bigcup_{i=1}^{k} R(q_i, \mathcal{D})$. This variant reflects the quality of reward-optimized fan-out but incurs the cost of autoregressive generation and repeated retrieval calls.

**R4T-Diffusion (RL-distilled diffusion retriever).** We distill the reward-aligned fan-out behavior of $\pi_{\theta^*}$ into a lightweight diffusion model $D_\phi$ trained on synthetic supervision (Section 2.5). At inference time, $D_\phi$ generates $L$ retrieval directions in a single non-autoregressive pass, $\mathbf{Z}_0 \sim p_\phi(\mathbf{Z} \mid z_q)$, which are mapped to database contents via nearest-neighbor retrieval. This variant evaluates whether reward-optimized fan-out can be retained under efficient single-pass inference.

Comparing these variants disentangles the benefits of reward-optimized fan-out from the cost of deploying it, directly testing whether RL-discovered behaviors can be distilled for efficient inference.

## Step 1: Fan-Out LM Training

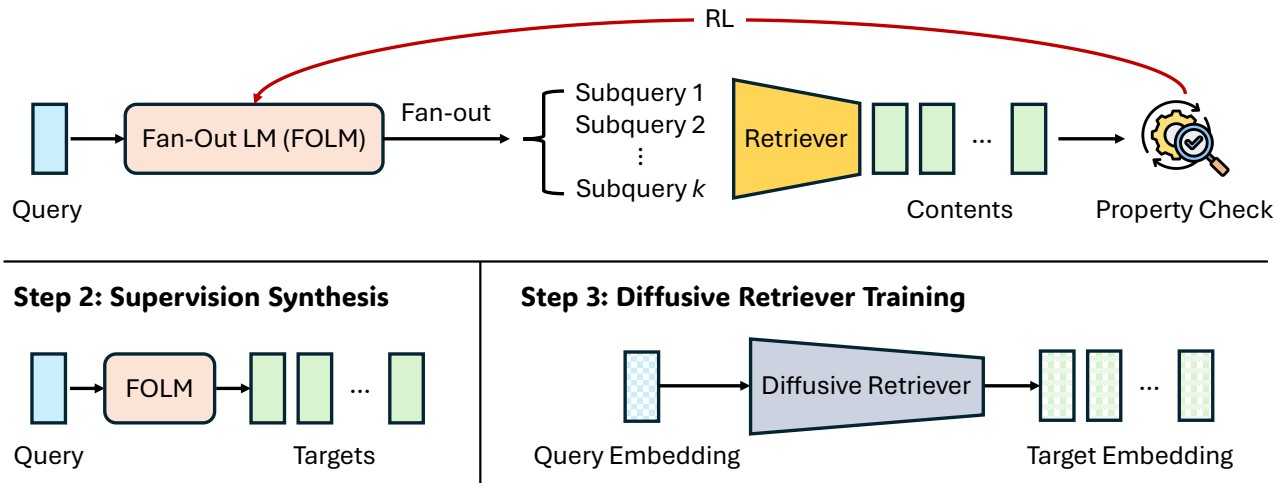

## Step 2: Supervision Synthesis

## Step 3: Diffusive Retriever Training

Figure 1. **Overview of the R4T.** Step 1 (§2.4) trains a fan-out language model (FOLM) using RL to produce property-aligned sub-queries. Step 2 (§2.5) uses the trained FOLM to synthesize $(q, c)$ supervision data. Step 3 (§2.6) trains a diffusion-based fan-out retriever that samples content embeddings directly from query embeddings.

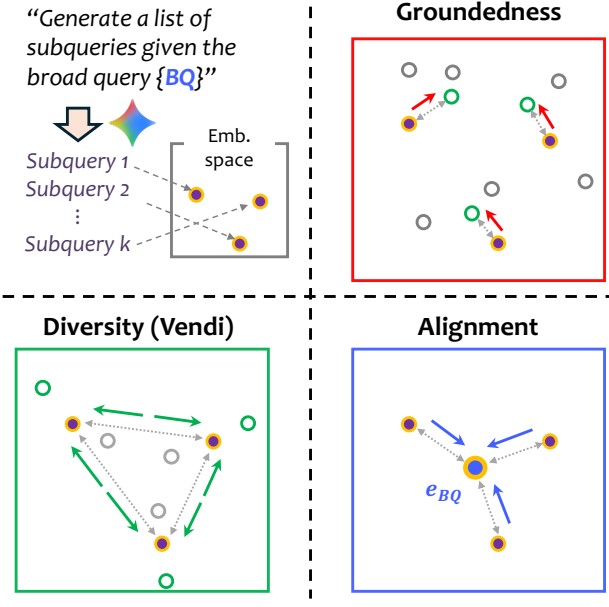

Figure 2. Illustration of rewards used in OAR.

### 2.3. Set-Level Objectives and Training Rewards

A key component of R4T is the ability to specify set-level objectives as rewards to be used in FOLM policy optimization (Stage 1). We study two regimes that capture common set-valued retrieval needs.

**(A) Open-Ended Abstract Retrieval (OAR).** OAR targets open-ended exploration where *no unique ground truth set exists*. Given a broad query $q$ (e.g., a theme or scenario), the goal is to retrieve a set of *collection-level* or abstract

results that are (i) diverse (i.e., cover multiple interpretations), (ii) well-aligned with $q$, and (iii) grounded with respect to the database.

We define a composite reward $\mathcal{R}_{\text{abs}}$ over a generated fan-out set $Q$ as follows:

$$\mathcal{R}_{\text{abs}}(q, Q) = \lambda_g r_{\text{ground}}(Q) + \lambda_d r_{\text{div}}(Q) + \lambda_a r_{\text{align}}(q, Q). \quad (1)$$

This reward uses subrewards $r_{\text{ground}}$, $r_{\text{div}}$ and $r_{\text{align}}$, to capture groundedness, diversity and alignment, respectively, each contributing according to their mixture weights ($\lambda$).

To capture diversity of the results set, we measure the semantic breadth of the fan-out using the Vendi Score (Friedman & Dieng, 2022) computed on representatives of each sub-query (e.g., top-1 retrieved item per sub-query), that is,

$$r_{\text{div}}(Q) = \text{Vendi}(\{e_{\text{content}}(c_i^\star)\}_{i=1}^k), \quad (2)$$

where $c_i^\star$ is a representative retrieved item for $q_i$ and $e_{\text{content}}(\cdot)$ is the content encoder that embeds database items.

To ensure groundedness, we encourage sub-queries to stay on the database manifold (induced by the embedding space) by penalizing the distance between each sub-query embedding and its nearest neighbor in the database, namely,

$$r_{\text{ground}}(Q) = 1 - \frac{1}{k} \sum_{i=1}^k \min_{c \in \mathcal{D}} \|e_{\text{text}}(q_i) - e_{\text{content}}(c)\|_2, \quad (3)$$

where $e_{\text{text}}(q_i)$ is a query embedding.

Finally, alignment is used to prevent semantic drift from the intent of the original query $q$ by anchoring each sub-query to $q$ with the following reward:

$$r_{\text{align}}(q, Q) = \frac{1}{k} \sum_{i=1}^{k} \cos(e_{\text{text}}(q_i), e_{\text{text}}(q)). \quad (4)$$

We set $\lambda_g = 0.6$ and $\lambda_d = \lambda_a = 0.2$ by default. Figure 2 illustrates the rewards used for OAR tasks.

**(B) Weakly Supervised Compositional Retrieval (WSCR).** WSCR targets settings where each query admits many valid item sets, but we have *weak reference sets* that represent one plausible realization. Each query $q$ is paired with a reference set $\mathcal{Y} = \{y_1, \ldots, y_m\}$; importantly, $\mathcal{Y}$ is not assumed to be the unique correct query response.

We define a reference-set-based coverage reward for a fan-out set $Q$ as follows:

$$\mathcal{R}_{\text{set}}(q, Q; \mathcal{Y}) = \frac{|\mathcal{Y} \cap \mathcal{R}(Q)|}{|\mathcal{Y}|}. \quad (5)$$

This reward encourages the policy to generate *complementary* sub-queries whose union spans different semantic components reflected in $\mathcal{Y}$, rather than memorizing a fixed target.

## 2.4. Fan-Out LM Training via Soft-GRPO

The first stage of R4T is RL training of the fan-out language model (FOLM). Given an input query $q$, the FOLM $\pi_\theta$ generates a set of candidate sub-queries $Q = \{q_1, \ldots, q_k\}$. These sub-queries are executed by a frozen dense retriever $R(\cdot)$ to obtain candidate documents $\mathcal{C}_i = R(q_i, \mathcal{D})$. We assign a reward to each sampled fan-out output using the relevant task objective (as defined in Section 2.3).

To optimize $\pi_\theta$, we employ *group relative policy optimization (GRPO)*. For a query $q$, we sample a set of $G$ outputs $\{o_1, \ldots, o_G\}$ and compute advantages using group statistics:

$$A_i = \frac{r_i - \mu_G}{\sigma_G + \epsilon}, \quad \mu_G = \frac{1}{G} \sum_{j=1}^{G} r_j. \quad (6)$$

**Soft-PPO Regularization.** To stabilize open-ended generation, we adopt *soft PPO* (Zhang et al., 2025b; Becker et al., 2025; Liang et al., 2025), which augments GRPO with both forward and reverse KL penalties between the active policy $\pi_\theta$ and the sampling policy $\pi_{\text{old}}$:

$$\mathcal{J}(\theta) = \mathbb{E}_{\pi_{\text{old}}} \big[ \mathcal{L}_{\text{GRPO}} - \beta_1 \mathbb{D}_{\text{KL}}(\pi_\theta \| \pi_{\text{old}}) \\ - \beta_2 \mathbb{D}_{\text{KL}}(\pi_{\text{old}} \| \pi_\theta) \big]. \quad (7)$$

In practice we implement this as:

$$\mathcal{L}(\theta) = \mathbb{E}_t \Big[ - \min(\rho_t A_t, \text{clip}(\rho_t, 1 - \epsilon, 1 + \epsilon) A_t) \\ + \beta_1 \cdot \rho_t (\log \pi_\theta(o_t) - \log \pi_{\text{old}}(o_t)) \quad (8) \\ + \beta_2 \cdot (-\log \pi_\theta(o_t)) \Big],$$

where $\rho_t = \pi_\theta / \pi_{\text{old}}$ is the importance ratio.

## 2.5. Synthetic Supervision

The second phase of R4T involves the generation of a synthetic dataset used to train the retrieval model. The FOLM $\pi_{\theta*}$ serves as a *behavior generator* that induces a distribution over fan-out retrieval patterns shaped by the reward. For each query $q$, we sample fan-out outputs from $\pi_{\theta*}$ and execute retrieval against the fixed database. We use the FOLM generations as supervision, allowing the downstream model to learn the full distribution of behaviors induced by the RL process above.

To train a downstream model that generates multiple retrieval embeddings in a single forward pass, we represent each fan-out result as a *coherent target tensor* $\mathbf{Z}_{\text{target}} \in \mathbb{R}^{L \times d}$. Each row of $\mathbf{Z}_{\text{target}}$ corresponds to one retrieval direction discovered by the FOLM. The construction of $\mathbf{Z}_{\text{target}}$ depends on the optimization objective:

**OAR (content-embedding targets).** For OAR tasks, the objective emphasizes the production of diverse and grounded *retrieved results*. Accordingly, we construct $\mathbf{Z}_{\text{target}}$ from the embeddings of the retrieved contents $\{z_{c_1}, \ldots, z_{c_L}\}$ corresponding to the fan-out outputs. This formulation directly distills the distribution over database-grounded collections induced by the RL-trained policy.

**WSCR (sub-query-embedding targets).** For WSCR tasks, the objective emphasizes discovery of complementary *search directions* that jointly cover a reference set. In this case, we construct $\mathbf{Z}_{\text{target}}$ from the embeddings of the optimized sub-queries generated by the FOLM, $\{e_{\text{text}}(q_1), \ldots, e_{\text{text}}(q_L)\}$. This choice allows the downstream model to internalize the search decomposition strategy learned by RL.

Because set-valued retrieval is order-agnostic, we randomly permute the rows of $\mathbf{Z}_{\text{target}}$ during training to encourage permutation robustness. The resulting synthetic dataset is $\mathcal{T}_{\text{syn}} = \{(z_q, \mathbf{Z}_{\text{target}})\}$, which captures the full fan-out retrieval distribution induced by the reward-optimized policy.

## 2.6. Diffusion for Single-Pass Fan-out

The final phase of R4T is the training of a generative retriever $D_\phi$ to model $p(\mathbf{Z}_{\text{target}} \mid z_q)$. We adopt the *variance exploding (VE) diffusion* formulation of Song et al. (2020) within the EDM framework (Karras et al., 2022). The de-

noiser $D_\phi(\mathbf{Z}_t; \sigma, z_q)$ is trained to recover clean targets from noisy inputs using the loss

$$\mathcal{L}_{\text{diff}} = \mathbb{E}_{\sigma,\epsilon} \left[ \lambda(\sigma) \cdot \| D_\phi(\mathbf{Z}_{\text{target}} + \sigma\epsilon; \sigma, z_q) - \mathbf{Z}_{\text{target}} \|^2 \right], \quad (9)$$

where $\lambda(\sigma) = (\sigma^2 + \sigma_{\text{data}}^2)/(\sigma \cdot \sigma_{\text{data}})^2$ balances contributions across noise levels.

**Architecture.** The denoiser is instantiated as a transformer (Peebles & Xie, 2023; Tomasi et al., 2025) tailored for structural coherence, adopting the preconditioning scheme proposed by Karras et al. (2022). The query embedding $z_q$ is injected via cross-attention. We use *classifier-free guidance (CFG)* (Ho & Salimans, 2022) by randomly dropping $z_q$ during training. At inference time, we solve the probability flow stochastic differential equation to generate $\mathbf{Z}_0$, which is sliced into $L$ embeddings and mapped to database contents via nearest-neighbor retrieval.

# 3. Experiments

## 3.1. Experimental Setup

We evaluate R4T on both OAR (no ground-truth; property-defined quality) and WSCR (weak reference sets; coverage-oriented) tasks, with objectives as defined in Section 2.3.

**Datasets.** We evaluate R4T on two diverse real-world datasets that reflect distinct retrieval modalities (text-to-image, text-to-music) and domains. The first is **Polyvore**, a large-scale fashion benchmark (Han et al., 2017) comprised of user-curated outfits. Each outfit functions as a ground-truth item set, containing compatible fashion products across diverse categories (e.g., tops, bottoms, shoes, accessories). The dataset provides rich multimodal features, including product images and textual metadata, which we use to evaluate both multimodal groundedness and set coherence. The candidate retrieval pool sizes are 21,888 (collections/abstracts) and 142,472 (items) for Task 1 and Task 2, respectively. **Music**, our second dataset, is a proprietary industrial dataset consisting of expert-generated music playlists. Each playlist represents a coherent sequence of tracks, serving as a ground-truth set for retrieval. This dataset evaluates the model's ability to model thematic consistency and intent coverage in the music domain. The candidate retrieval pool size is 8,522 (playlist embeddings) for Task 1. Details of broad query generation are provided in Appendix D.

**Retrieval Backbone.** We adopt dataset-specific embedding backbones to enable efficient fan-out in a shared embedding space. For Polyvore, we use a CLIP-based image-text encoder trained with matryoshka representation learning (Kusupati et al., 2022), which supports multimodal retrieval while allowing flexible embedding truncation. Un-

less otherwise specified, we use an embedding dimension of 128 to balance retrieval accuracy and efficiency in our main experiments. For the music dataset, we employ Mu-Lan (Huang et al., 2022), a joint music–text embedding model trained on large-scale audio-language pairs, and perform retrieval directly in the MuLan embedding space.

**Baselines** We compare R4T against three baselines. The **No Fan-out** baseline directly retrieves $n \times k$ content items using the original query without any sub-query expansion. This represents the traditional dense retrieval approach and serves as a lower bound for fan-out methods. A second baseline, **Zero-shot Fan-out** uses the base language model (before RL training) to generate $k$ sub-queries without any task-specific optimization. Each sub-query retrieves $n$ items, resulting in $n \times k$ total items. We test several variants of the Zero-shot Fan-out baseline with different language models used for sub-query generation: (a) **Gemini-2.5-Flash** (Comanici et al., 2025), a large-scale proprietary model with strong zero-shot capabilities; (b) **Gemma3-4B** (Team et al., 2025), a smaller open-source model (4B parameters); and (c) **Qwen3-4B** (Yang et al., 2025), another competitive open-source model (4B parameters). **Best-of-N** is the third baseline, a strong baseline that performs fan-out $N$ times using the zero-shot model and selects the best result according to our training rewards. We set $N = 5$.

We compare two variants of our method, **R4T-(FOLM/Diffusion)**. R4T first trains the fan-out language model using RL with task-specific rewards, synthesizes training data, and trains a diffusion-based retriever. At inference time, the diffusion retriever directly samples $k$ content embeddings from the query embedding. For all fan-out baselines and R4T, we report results with $k = 10$ sub-queries.

**Evaluation Metrics.** We evaluate set-valued retrieval using task-specific metrics. For OAR tasks, where no ground-truth sets exist, we employ LLM-as-a-Judge evaluation using Gemini-2.5-Pro to assess three dimensions: *collection diversity*, *query-collection alignment*, and *groundedness*. Each dimension is scored on a 5-point Likert scale, following prior work showing strong correlation with human judgments (Zheng et al., 2023).

For WSCR tasks, we report reference-based coverage metrics (Recall@5K and Hit@5K) together with the Vendi Score to measure diversity and generation stability. Since reference sets represent one plausible realization of the query intent rather than exhaustive ground truth, recall is interpreted as a proxy for semantic coverage rather than binary correctness.

Full evaluation protocols, prompt templates, and metric definitions are provided in Appendices C, E, and F.

| | Polyvore | | | | Music | | | |
|---|---|---|---|---|---|---|---|---|
| | Groundedness | Diversity | Alignment | Average | Groundedness | Diversity | Alignment | Average |
| No Fan-out | 22.4 | 34.4 | 21.4 | 26.1 | 48.8 | 20.0 | 41.8 | 36.9 |
| **Gemini-2.5-Flash** | | | | | | | | |
| Zero-shot | 24.0 | 47.0 | 23.6 | 31.5 | 45.8 | 45.2 | 44.4 | 45.1 |
| Best-of-N | $26.1_{\pm1.6}$ | $52.2_{\pm2.3}$ | $25.2_{\pm1.9}$ | $34.5_{\pm1.8}$ | $48.2_{\pm2.1}$ | $48.4_{\pm2.6}$ | $49.0_{\pm2.0}$ | $48.5_{\pm2.2}$ |
| **Gemma3-4B** | | | | | | | | |
| Zero-shot | 28.4 | 56.0 | 31.2 | 38.5 | 49.8 | 42.6 | 51.8 | 48.1 |
| Best-of-N | $28.9_{\pm1.5}$ | $61.0_{\pm2.1}$ | $32.7_{\pm1.7}$ | $40.9_{\pm1.6}$ | $51.4_{\pm2.0}$ | $43.2_{\pm2.3}$ | $53.0_{\pm1.9}$ | $49.2_{\pm2.1}$ |
| R4T-FOLM | $\mathbf{30.8_{\pm1.9}}$ | $\mathbf{76.8_{\pm2.7}}$ | $\mathbf{39.8_{\pm2.3}}$ | $49.1_{\pm2.1}$ | $\mathbf{63.1_{\pm2.2}}$ | $\mathbf{49.2_{\pm2.4}}$ | $\mathbf{62.0_{\pm2.0}}$ | $58.1_{\pm2.2}$ |
| R4T-Diffusion | \ | $74.3_{\pm3.4}$ | $37.6_{\pm2.8}$ | \ | \ | $46.7_{\pm3.1}$ | $59.6_{\pm2.6}$ | \ |
| **Qwen3-4B** | | | | | | | | |
| Zero-shot | 23.8 | 37.0 | 23.4 | 28.1 | 42.0 | 38.8 | 41.2 | 40.7 |
| Best-of-N | $27.0_{\pm1.8}$ | $40.3_{\pm2.5}$ | $24.0_{\pm1.9}$ | $30.4_{\pm1.7}$ | $44.0_{\pm2.2}$ | $40.3_{\pm2.7}$ | $43.7_{\pm2.1}$ | $42.7_{\pm2.3}$ |
| R4T-FOLM | $\mathbf{37.0_{\pm2.1}}$ | $62.8_{\pm2.9}$ | $\mathbf{28.0_{\pm2.0}}$ | $42.6_{\pm2.2}$ | $\mathbf{48.2_{\pm2.3}}$ | $\mathbf{44.8_{\pm2.5}}$ | $49.4_{\pm2.1}$ | $47.5_{\pm2.2}$ |
| R4T-Diffusion | \ | $\mathbf{65.0_{\pm3.6}}$ | $27.4_{\pm2.7}$ | \ | \ | $44.5_{\pm3.0}$ | $\mathbf{52.0_{\pm2.8}}$ | \ |

*Table 1.* **Performance on Task 1: Open-Ended Abstract Retrieval (OAR).** Results are reported as mean $\pm$ standard deviation. **Bold** denotes the best performance within each model family. R4T consistently outperforms fan-out baselines across datasets and metrics. Note that Groundedness is not applicable to R4T-Diffusion (indicated by "\") due to the absence of intermediate sub-query.

| | Recall@5K | Hit@5K | VS |
|---|---|---|---|
| **LLM-Based Retrieval** | | | |
| Gemini-2.5-Flash | 15.7 | 52.1 | 33.4 |
| Gemma3-4B | 6.0 | 25.9 | 44.2 |
| Qwen3-4B | 10.1 | 33.9 | 46.4 |
| R4T-FOLM (Gemma) | 16.9 | 54.4 | 40.5 |
| R4T-FOLM (Qwen) | **20.9** | **64.6** | **27.5** |
| **Diffusion-Based Retrieval** | | | |
| R4T-Diffusion (Gemma) | 15.0 | 54.1 | 46.2 |
| R4T-Diffusion (Qwen) | **16.5** | **57.5** | **34.7** |

*Table 2.* **Performance on Task 2: Weakly Supervised Compositional Retrieval (WSCR).** We use an LLM (Gemini-2.5-Pro) to generate a broad query for each outfit in Polyvore, and set the items in each as the ground truths. For diversity evaluation with normalized Vendi Score, we set temperature=0.9 for LLM-based retrieval, and set #samples=5 (i.e., 5×10=50 sub-queries in total) for all the methods.

### 3.2. Main Results

Table 1 and Table 2 summarize the performance of all methods on the OAR and WSCR tasks. We highlight several consistent patterns that clarify both the strengths and limitations of existing fan-out strategies, as well as the advantages of R4T.

**Fan-out is necessary but not sufficient.** Across all datasets and base models, zero-shot fan-out consistently outperforms the No Fan-out baseline that retrieves directly using the original query. This confirms the long-standing hypothesis that query expansion improves coverage of di-

verse semantic facets (Carpineto & Romano, 2012). However, zero-shot fan-out exhibits clear weaknesses. Generated sub-queries often drift off the database manifold or collapse into near-duplicate expressions, leading to limited groundedness and redundant retrieval results. These effects are particularly visible in the OAR task, where no ground truth constrains exploration.

**Reward-aware selection improves quality but does not scale.** The Best-of-N baseline substantially improves over zero-shot fan-out by selecting high-reward fan-out instances. This demonstrates that the reward functions we design are meaningful signals for shaping retrieval behavior. However, Best-of-N requires multiple independent fan-out executions per query, resulting in an order-of-magnitude increase in inference cost. As a result, its gains come at the expense of latency and scalability, limiting its applicability in practical systems.

**R4T consistently dominates the accuracy-efficiency frontier.** R4T achieves the strongest overall performance across datasets, tasks, and base models. Notably, R4T outperforms Best-of-N while requiring only a single inference pass at deployment time. This result highlights the core advantage of R4T: reinforcement learning is used to discover high-quality fan-out behaviors once, and these behaviors are then distilled into a lightweight diffusion-based retriever. The RL-trained FOLM learns to generate sub-queries that balance diversity, alignment, and groundedness, while the diffusion model faithfully captures this distribution in embedding space, enabling efficient and controllable fan-out retrieval.

| Model | Sub Query | Retrieved Images |
|---|---|---|
| R4T | bohemian festival dress |  |
| R4T | straw boots festival style |  |
| R4T | lace bohemian festival |  |
| Qwen3_4b | bohemian festival style |  |
| Qwen3_4b | bohemian festival fashion |  |
| Qwen3_4b | festival bohemian clothes |  |

*Figure 3.* Qualitative comparison on the Polyvore dataset for the open-ended abstract retrieval task given the broad query **"Bohemian festival style"**. R4T generates semantically distinct, on-topic sub-queries that retrieve diverse outfit collections, while the Qwen3-4B zero-shot baseline produces largely paraphrastic sub-queries, leading to more homogeneous results.

**Reward Hacking in OAR.** Careful reward design is crucial for realizing R4T's performance gains. We perform an ablation on the OAR task (Figure 4) to analyze how different reward components shape fan-out behavior. Without a diversity term, the reward is easily exploited. Training with only the *groundedness* reward, the policy converges to degenerate, nonsensical strings (e.g., *"line ending line ending line ending"*), which happen to minimize embedding distance to a specific database item. When trained with both *groundedness* and *alignment*, collapse occurs even faster, as the policy simply repeats paraphrases of the original query, trivially maximizing alignment while ignoring semantic dispersion. By contrast, jointly optimizing groundedness, alignment, and diversity prevents such collapse and yields stable GRPO training.

Figures 4 (d–f) further illustrate how reward weighting affects training dynamics. When groundedness dominates, diversity increases but alignment degrades, indicating an overemphasis on database proximity at the expense of semantic fidelity. Conversely, emphasizing alignment and diversity suppresses exploration, limiting coverage across alternative interpretations or intents. A balanced weighting yields stable convergence of all three scores, highlighting an inherent trade-off between exploration and semantic fidelity. Together, these results show that diversity and alignment act as mutual counter-anchors, forcing the policy into a balanced region of the reward landscape where shortcut solutions are ineffective and high reward can only be achieved by generating meaningful, database-grounded sub-query variations.

**Set Retrieval Behavior.** Table 2 reveals a clear trade-off between reference-based coverage and diversity across retrieval paradigms. Among LLM-based methods, higher Recall@5K and Hit@5K are often accompanied by lower Vendi Scores, indicating that autoregressive fan-out tends to concentrate retrieval around a small number of dominant semantic modes when optimized for reference overlap. This behavior is consistent with recent findings on the use of RL for language models, which show that stronger reward optimization typically reduces output entropy and

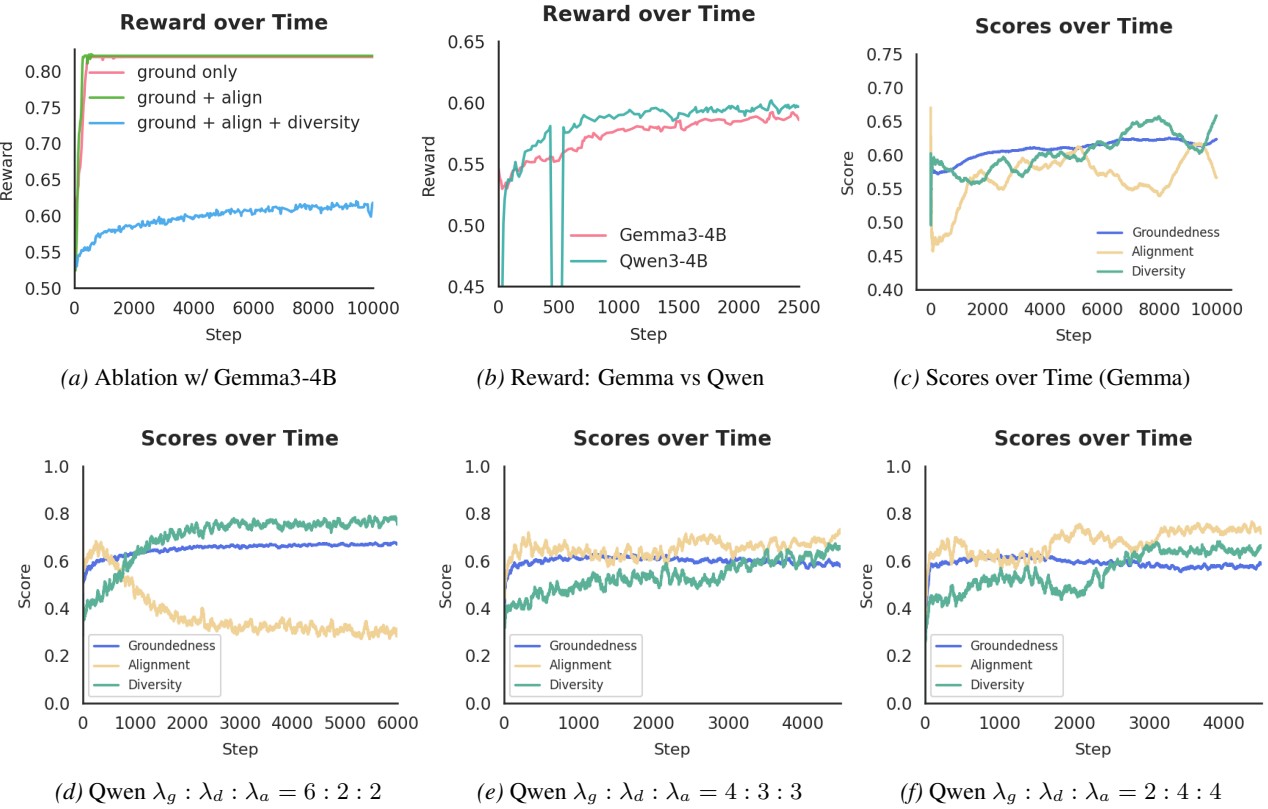

*(a)* Ablation w/ Gemma3-4B     *(b)* Reward: Gemma vs Qwen     *(c)* Scores over Time (Gemma)

*(d)* Qwen $\lambda_g : \lambda_d : \lambda_a = 6 : 2 : 2$     *(e)* Qwen $\lambda_g : \lambda_d : \lambda_a = 4 : 3 : 3$     *(f)* Qwen $\lambda_g : \lambda_d : \lambda_a = 2 : 4 : 4$

*Figure 4.* Reward & Scores over Time (FOLM Training) for Open-Ended Abstract Retrieval.

encourages mode collapse (Cui et al., 2025). By contrast, zero-shot LLM baselines and diffusion-based retrieval exhibit higher diversity, reflecting broader exploration of the semantic space across inference runs.

Importantly, lower recall in these cases does not necessarily imply lower retrieval quality. The reference sets do not reflect an "exhaustive ground" truth, but rather represent only just one (of potentially many) plausible realization of the query intent. Greater diversity therefore suggests that the model may be uncovering alternative, equally valid interpretations that are not captured by the reference items. R4T shifts this trade-off in a favorable direction. While R4T-FOLM improves reference coverage at the cost of reduced diversity, R4T-Diffusion preserves much of the diversity inherent to diffusion-based generation while substantially improving coverage. This indicates that distilling reward-aligned behaviors into a diffusion prior offers a practical way to balance coverage and semantic richness for set-valued retrieval under weak supervision.

### 3.3. Qualitative Example

Figure 3 and Figure 6 show qualitative results from the OAR task on Polyvore for two broad queries, *"Bohemian festival style"* and *"Labor day picnic outfit"*. These ex-

amples illustrate how the joint use of groundedness, alignment, and diversity rewards guides R4T to produce meaningful and varied sub-query decompositions. For the *Bohemian festival style* query, R4T generates sub-queries that branch into distinct thematic directions such as *"bohemian festival dress"*, *"straw boots festival style"*, and *"lace bohemian festival"*. Each sub-query retrieves a visually coherent yet semantically distinct group of items, indicating strong diversity while remaining consistent with the overall style. In contrast, baseline models such as Qwen-4B often produce near-synonymous variations of the same expression, for example *"bohemian festival style"* and *"bohemian festival fashion"*, which leads to more homogeneous retrieval results. A similar trend is observed for the *"Labor day picnic outfit"* query. R4T decomposes the query into complementary facets such as *bohemian*, *minimalist*, and *jumpsuit* styles, retrieving outfits that differ in silhouette, color palette, and accessory choices while remaining appropriate for the seasonal picnic context. Gemini-2.5-Flash, although capable of generating syntactically diverse sub-queries, tends to retrieve sets that are more uniform and exhibit limited semantic coverage.

Overall, these qualitative results highlight the role of the three reward components in shaping retrieval behavior. Groundedness encourages each sub-query to correspond to

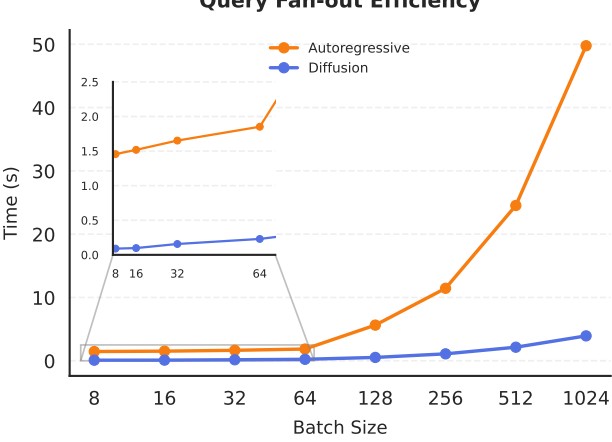

**Query Fan-out Efficiency**

*Figure 5.* **Efficiency comparison between Autoregressive LLM and Diffusion Model for query fan-out ($k = 10$) generation.** Note that the x-axis follows a logarithmic scale (doubling at each step). The Diffusion Model demonstrates superior scalability and lower latency, maintaining sub-second performance for small batches and achieving an order-of-magnitude speedup at larger batch sizes.

real items in the database, alignment preserves consistency with the original query intent, and diversity promotes exploration of multiple valid interpretations. Together, these objectives allow R4T to generate rich query decompositions that translate into diverse and relevant retrieval.

### 3.4. Efficiency Analysis

To evaluate inference-time efficiency, we benchmark the latency of query fan-out for R4T against autoregressive LLM-based fan-out baselines. We compare a diffusion-based retriever that generates all $k = 10$ retrieval directions in a single forward pass with autoregressive models that sequentially generate sub-queries and invoke retrieval. Wall-clock latency is measured across varying batch sizes.

Figure 5 illustrates the wall-clock time required for fan-out generation. The autoregressive LLM is dominated by high constant overheads at smaller batch sizes, taking approximately 1.46 seconds even for a batch of 8, before scaling linearly to nearly 50 seconds at a batch size of 1024. In contrast, our diffusion model, containing only 53.9M parameters, leverages non-autoregressive generation in the embedding space to process the small batch in just 0.07 seconds and the largest batch in 4.21 seconds. This compact architecture drastically reduces deployment memory overhead while delivering a consistent $12\times$-$20\times$ speedup , confirming that transforming the fan-out burden from a heavy autoregressive System 2 to a lightweight System 1 diffusion prior is essential for practical, real-time retrieval applications.

## 4. Conclusion

We present R4T, a reinforcement learning-based framework for synthesizing training data for generative retrieval. By training a fan-out language model with property-specific rewards and using it to generate supervision for a diffusion-based retriever, we enable controllable fan-out retrieval without human-labeled data. Experiments on fashion product benchmark demonstrate that R4T outperforms strong baselines while maintaining efficient inference. Our work opens new directions for training generative retrieval systems in specialized domains where supervision is scarce. Related work and limitations are discussed in Appendix A and B, respectively.

## Impact Statement

This work addresses the growing gap between increasingly rich retrieval objectives and the limited supervision available to train efficient retrieval systems under those objectives. By using reinforcement learning as an objective transducer and compiling the resulting behaviors into a lightweight diffusion-based retriever, R4T enables controllable, set-valued retrieval without requiring human-labeled data. This design has several potential positive impacts.

From a systems perspective, R4T provides a practical pathway for deploying retrieval models that optimize higher-order properties such as diversity, coverage, and complementarity while maintaining low inference latency. This is particularly relevant for real-world applications where fan-out retrieval is desirable but autoregressive generation is prohibitively expensive, including recommendation systems, creative search, and exploratory information access. By separating reward-driven discovery from inference-time deployment, our framework supports scalable and customizable retrieval without repeated online optimization.

From a research perspective, this work contributes a general paradigm for transforming non-decomposable, set-level objectives into trainable supervision. ***The idea of using RL to synthesize objective-aligned training data may extend beyond retrieval to other structured generation tasks where ground truth is ambiguous or subjective, such as planning, design, and creative generation.*** We hope this encourages further exploration of compiled approaches that combine interactive learning with efficient generative models.

**Ethical Considerations and Societal Impact.** Because retrieval behaviors are shaped by explicitly designed reward functions, careless reward specification could encode or amplify biases present in training data, potentially leading to unfair representation in applications like recommendation systems or content discovery. The RL-based synthesis approach may propagate such biases at scale through syn-

thetic training data. While our experiments focus on benign domains (fashion, music), the same techniques could be applied to sensitive contexts where retrieval biases may have significant consequences. Responsible deployment requires domain-specific bias audits, inclusive design practices, and appropriate oversight mechanisms. We view R4T as a tool for controlled retrieval design that must be accompanied by safeguards rather than a substitute for human judgment and ethical oversight.

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

# A. Related Work

**Generative Retrieval.** Recent work has explored formulating retrieval as a generation task. DSI (Tay et al., 2022) trains sequence-to-sequence models to generate document identifiers (docids) directly from queries, treating the entire corpus as model memory. NCI (Wang et al., 2022) extends this with improved indexing strategies, while SEAL (Bevilacqua et al., 2022) generates n-grams for passage retrieval. More recent work has explored semantic identifiers (Li et al., 2023), learning to rank in generative retrieval (Li et al., 2024), and scalable approaches for large corpora (Pradeep et al., 2023; Zeng et al., 2024). Our work differs by generating in embedding space rather than discrete token space, enabling smoother interpolation and multimodal retrieval. Unlike prior generative retrieval systems that require extensive human-labeled supervision, we synthesize training data through RL.

**Query Expansion and Reformulation.** Query expansion has a long history in information retrieval (Carpineto & Romano, 2012; Azad & Deepak, 2019), with classic techniques including pseudo-relevance feedback (Rocchio, 1971) and term co-occurrence analysis. Modern approaches leverage neural language models for query reformulation (Nogueira et al., 2019), with recent work exploring LLM-based query generation (Ma et al., 2023) and multi-query decomposition for complex information needs. However, these methods typically operate in a two-stage retrieve-then-rerank pipeline and do not explicitly ground expansions in the target database. Our fan-out approach differs by training query expansion to optimize database-specific retrieval properties through reinforcement learning.

**Reinforcement Learning for Retrieval.** Recent work has begun applying RL to information retrieval tasks. DeepRetrieval (Jiang et al., 2025a) and s3 (Jiang et al., 2025c) use RL to train search agents that interact with real search engines, demonstrating that reward-driven optimization can encode fine-grained retrieval objectives. Zhou et al. (2023) explores RL from relevance feedback for generative retrieval. However, these methods use RL for direct inference, incurring high computational costs at query time. We instead use RL as a *data generation engine*, amortizing the cost across inference queries by synthesizing training data once and then deploying an efficient diffusion-based retriever.

**Diffusion Models for Retrieval.** Diffusion probabilistic models (Ho et al., 2020; Song et al., 2020) have shown strong performance in generative modeling across multiple domains. Recent work has adapted diffusion models to retrieval tasks. Diff4Steer (Bao et al., 2025) applies diffusion priors to music retrieval with semantic guidance, while GDRetriever (Guinot et al., 2025) develops controllable

generative text-music retrieval. These works demonstrate that sampling in embedding space enables flexible multi-hypothesis generation. Our work extends this paradigm by addressing the training data bottleneck: rather than relying on human supervision, we synthesize property-aligned training data through RL-guided generation.

**Synthetic Data Generation.** Synthetic data generation has emerged as a powerful technique for addressing data scarcity and privacy concerns in machine learning (Lu et al., 2023; Goyal & Mahmoud, 2024). Techniques range from rule-based systems to deep generative models including GANs (Goodfellow et al., 2014) and VAEs (Kingma & Welling, 2013). Recent work has explored using LLMs to generate training data for various tasks (Chen et al., 2023), including instruction tuning and task-specific fine-tuning. In retrieval, synthetic queries have been generated for training dense retrievers (Ma et al., 2021), but typically without property-specific control. Our work uniquely combines RL-based data synthesis with diffusion-based retrieval, enabling controllable generation of property-aligned training pairs.

**Multimodal Retrieval.** Cross-modal retrieval systems (Wang et al., 2025) aim to bridge semantic gaps between different modalities such as text, images, and audio. Vision-language models like CLIP (Radford et al., 2021) and CLAP (Wu et al., 2023) learn joint embedding spaces for retrieval. Recent work has explored generative approaches for multimodal retrieval (Wu et al., 2024), including retrieval-augmented generation for multimodal tasks. Our framework operates on top of frozen multimodal encoders, demonstrating that RL-guided data synthesis can enhance retrieval across modalities without modifying the underlying embedding space.

# B. Limitations

Despite its effectiveness, R4T has several limitations that suggest directions for future work.

First, the reinforcement learning stage requires repeated interaction with a frozen retriever and explicit reward computation. While this cost is amortized at deployment time, the upfront training overhead may be substantial for extremely large or frequently changing databases. Future work could explore more sample-efficient RL methods, partial retriever updates, or offline approximations to reduce this cost.

Second, R4T assumes that desired retrieval properties can be reasonably expressed as explicit reward functions. Although we show that combining groundedness, alignment, and diversity yields stable behavior, some user preferences, such as subjective notions of creativity, novelty, or cultural sensitivity, may be difficult to encode in scalar rewards. Learning reward functions from user feedback or prefer-

ence data is a promising direction to address this limitation.

Third, our evaluation relies in part on LLM-as-a-Judge for assessing open-ended retrieval quality. While prior work suggests strong correlation with human judgments, such evaluations may still introduce biases inherited from the judge model and may not fully capture all qualitative aspects of retrieval usefulness. Incorporating human evaluation or hybrid evaluation protocols would strengthen future studies.

Finally, our instantiation of R4T focuses on a specific fan-out architecture and diffusion-based retriever. Although the framework is general, empirical results may depend on the choice of base language model, embedding space, and diffusion architecture. Extending R4T to other retrieval backbones, modalities, and task settings remains an open area for exploration.

Overall, we view R4T as an initial step toward scalable training of set-valued retrieval systems under complex objectives, rather than a complete solution to all forms of controllable retrieval.

# C. Evaluation Metrics

## C.1. LLM-as-a-Judge Evaluation

For Open-Ended Abstract Retrieval (OAR), retrieval quality is defined by set-level properties that do not admit unique ground-truth targets. We therefore adopt LLM-as-a-Judge evaluation using Gemini-2.5-Flash, which supports multimodal inputs (text, images, and audio). This evaluation paradigm has been shown to correlate well with human judgments for retrieval quality assessment (Zheng et al., 2023).

We evaluate three dimensions:

1. **Collection Diversity:** The degree to which retrieved collections span distinct semantic interpretations of the broad query.

2. **Query–Collection Alignment:** How well the retrieved collections collectively reflect the intent of the original query.

3. **Groundedness:** How accurately each retrieved collection corresponds to its generating sub-query.

For each dimension, the LLM judge assigns a score on a 5-point Likert scale (1=Poor, 2=Fair, 3=Good, 4=Very Good, 5=Excellent), along with a brief justification. To reduce evaluation bias, we randomize item order, apply explicit evaluation criteria, require explanations for each score, and avoid any mention of the retrieval method in the prompts. Complete prompt templates are provided in Appendix F.

## C.2. WSCR Metrics

For Weakly Supervised Compositional Retrieval (WSCR), we evaluate retrieval quality using reference-based coverage metrics and a diversity measure. Each query is associated with a reference item set that represents one plausible realization of the intended semantics, rather than an exhaustive or uniquely correct target. Accordingly, recall-based metrics are interpreted as proxies for semantic coverage rather than strict correctness.

**Recall@5K.** Recall@5K measures the fraction of reference items retrieved within the candidate pool. We retrieve 500 candidates for each of the $k = 10$ sub-queries (or generated embeddings), yielding a pool of 5,000 items per inference pass.

**Hit@5K.** Hit@5K measures the percentage of queries for which at least one reference item appears in the retrieved candidate pool, capturing whether any semantic facet of the reference realization is recovered.

**Vendi Score.** To assess diversity and generation stability, we perform $N = 5$ independent inference runs per query. For each run, we compute a representative embedding by averaging the constituent sub-query embeddings. The Vendi Score (Friedman & Dieng, 2022) is computed across these embeddings to quantify semantic variance across runs.

# D. Broad Query Generation

To evaluate **R4T** across diverse retrieval regimes, we construct two types of synthetic broad queries corresponding to the two tasks studied in this paper: (1) **Open-Ended Abstract Retrieval (OAR)** and (2) **Weakly Supervised Compositional Retrieval (WSCR)**. This appendix describes the query construction procedures used for each task.

## D.1. OAR: Broad Query Generation

Broad queries for OAR are open-ended, exploratory textual prompts that describe general themes, styles, or scenarios without specifying concrete items. These queries are designed to elicit diverse, collection-level retrieval outputs that admit multiple valid interpretations, consistent with the absence of ground-truth supervision in OAR.

We adopt a multi-template generation strategy to ensure broad coverage of semantic intents and query styles. Queries are generated using a large language model with diverse prompt templates targeting different aspects of abstraction, including: aesthetic themes (e.g., "bohemian festival style"), activity-based scenarios (e.g., "weekend brunch outfit"), seasonal contexts (e.g., "summer vaca-

tion vibes"), lifestyle identities (e.g., "sustainable living"), mood expressions (e.g., "confident and bold"), cultural trends (e.g., "cottagecore aesthetic"), functional needs (e.g., "travel essentials"), and hybrid combinations (e.g., "vintage meets modern").

---

**Broad Query Generation Prompt for OAR**

**System:** You are a creative query generator for a retrieval system.
**User:** Generate 100 diverse, creative broad queries for a fashion/music retrieval system. Each query should be 2–6 words describing a general theme, aesthetic, or scenario.
Requirements:

- Queries should be open-ended and exploratory

- Focus on high-level themes rather than specific items

- Use natural, conversational language

- Avoid brand names or concrete product references

- Each query should admit multiple valid interpretations

Return only a JSON array of query strings.

---

After generation and deduplication, we obtain **43,874 unique broad queries** for the OAR task. These queries are randomly split into train/validation/test sets with an 8:1:1 ratio. For each query, we generate training supervision using the Fan-Out Language Model (FOLM) with temperature 0.9 and a sample size of 128 to synthesize targets for diffusion model training.

### D.2. Broad Query Generation for WSCR

Broad queries for WSCR are constructed in a fundamentally different manner from OAR. Instead of being generated independently, each WSCR query is *derived from an existing item set* and serves as a weak, reference-based description of one plausible compositional realization of a user intent.

Concretely, we start from curated item sets (e.g., outfits in Polyvore), each consisting of multiple complementary items. Given the item names within a set, we prompt a large language model to generate a concise, outfit-level query that captures the *overall style, aesthetic, or occasion* of the set, without enumerating individual items. The resulting query functions as a high-level semantic abstraction of the item composition, rather than a uniquely correct label.

This process yields pairs of the form $(q, \mathcal{Y})$, where $q$ is a broad query and $\mathcal{Y}$ is one plausible reference item set. Importantly, $\mathcal{Y}$ is *not* treated as ground truth: many alternative item sets may equally satisfy the semantics of $q$. These

query–set pairs therefore constitute weak supervision suitable for WSCR.

---

**WSCR: Broad Query Generation from Item Set**

**System:** You are a query generator for outfit-level retrieval.
**User:** Given the following fashion items in an outfit, generate a broad outfit-level search query that describes the overall style or theme of the outfit.
The query should be:

- 2–8 words

- Broad and general (do not list specific item details)

- Focused on overall style, aesthetic, or occasion

- Suitable for searching for similar outfits

Items:
```
– [item name 1]
– [item name 2]
...
```
Return **only** the query text.

---

**Training and Test Set Augmentation.**

---

**WSCR: Item-Set Augmentation and Query Generation**

**System:** You are an outfit composer and query generator.
**User:** Given a list of fashion items, select multiple sets of exactly 10 items each. Each set should form a coherent outfit with a distinct style. For each set, generate a broad outfit-level query (2–8 words) describing the overall style or occasion.
**Constraints:**

- Each set must contain exactly 10 items

- Sets should differ in style or occasion

- Queries should be broad and not enumerate item details

Return the selected item indices and the corresponding queries in a structured format.

---

To increase coverage and reduce overfitting to specific reference realizations, we further augment the WSCR data using structured item-set recomposition guided by a language model.

- **Training-set augmentation.** For each original item set, we construct a candidate pool of 50 items consisting of the original set plus randomly sampled items from the global item pool. A language model is prompted *in a single call* to select multiple (8) distinct subsets of 10 items, each forming a coherent outfit

with a different style. For each subset, a corresponding broad query is generated.

- **Test-set augmentation.** For each test item set, we sample multiple seed subsets containing 3–60% of the original items. For each seed, a language model selects additional items from a 30-item candidate pool to form complete 10-item outfits, each accompanied by a broad query. This procedure produces multiple plausible compositional realizations for a single query while preserving partial overlap with the original set.

Across all splits, this procedure yields **84,704 unique broad queries** for WSCR. The resulting dataset is split into train/validation/test sets using an 8:1:1 ratio. As in OAR, diffusion model training targets are generated by running the Fan-Out Language Model (FOLM) with temperature 0.9 and a sample size of 128 per query over the training set.

Overall, this construction ensures that WSCR evaluates retrieval under weak, reference-based supervision, where success is measured by *compositional coverage* rather than exact set matching.

## E. Query Fan-out Prompts

> **Query Fan-out for Open-Ended Abstract Retrieval (OAR)**
>
> You are a Query Writer. Given a broad query, your task is to create a list of queries which are diverse but cover the same topic as the broad query (better adding/changing one to three words). These queries will be used to search a fashion dataset.
>
> You should show your thinking process in `<think>` `</think>` tags. You MUST return the final list of queries in JSON format in `<queries>` `</queries>` tags.
>
> For example:
> ```
> <think>
> [thinking process]
> </think>
> <queries>
> [
>     "query1",
>     "query2",
>     "query3",
>     ... (up to 10 queries)
> ]
> </queries>
> ```
> The broad query is:
> ```
> <broad_query>
> {broad_query}
> </broad_query>
> ```

> **Query Fan-out for Weakly Supervised Compositional Retrieval (WSCR)**
>
> You are a Query Writer. Your task is to create a list of queries for a broad query given by me to retrieve a set of items from a fashion dataset to form an outfit.
>
> You should show your thinking process in `<think>` `</think>` tags. You MUST return the final list of queries in JSON format in `<queries>` `</queries>` tags.
>
> For example:
> ```
> <think>
> [thinking process]
> </think>
> <queries>
> [
>     "query1",
>     "query2",
>     "query3",
>     ... (up to 10 queries)
> ]
> </queries>.
> ```
> The broad query is:
> ```
> <broad_query>
> {broad_query}
> </broad_query>
> ```

## F. LLM Judge Evaluation Prompts

In this section, we provide the detailed prompts used for LLM-as-a-Judge evaluation. Our prompts are carefully designed to minimize bias and maximize fairness through several strategies:

1. **Randomization:** Items are presented in random order to avoid position bias

2. **Objective Criteria:** We provide clear, measurable evaluation dimensions

3. **Explanation Requirement:** The judge must provide reasoning for scores

4. **Method Agnostic:** No mention of retrieval methods or systems

5. **Calibration Examples:** We include anchor examples for score calibration

6. **Structured Output:** JSON format ensures consistent score extraction

## G. Implementation Details

**Fan-Out LM (FOLM).** We initialize the Fan-Out LM using a standard instruction-tuned checkpoint. Training utilizes the GRPO algorithm with Soft-PPO regularization,

---

**OAR - Collection Diversity Evaluation**

**System Prompt:**
You are an expert evaluator assessing the diversity of retrieved content items. You will be shown a set of items (product images or text descriptions) and asked to evaluate how diverse they are in terms of semantic content, themes, and categories.

**User Prompt:**
```
Original Query: {query}
Retrieved Items: [Images/text are presented in random order]
```

Please evaluate the **diversity** of this retrieved set on a scale of 1-5:

**Evaluation Criteria:**

- **5 (Excellent):** Excellent diversity. The set of [Image/audio] provides a comprehensive and creative exploration of the query with minimal redundancy.

- **4 (Very Good):** Strong variety. The [Image/audio] are mostly distinct and cover different facets of the query well.

- **3 (Good):** Decent variety. The [Image/audio] explore several different ideas or aspects of the query. Some overlap is present but acceptable. (This is the expected 'good' result).

- **2 (Fair):** Some repetition. The [Image/audio] show 2-3 distinct ideas, but many are redundant.

- **1 (Poor):** Highly redundant. The [Image/audio] are all very similar, exploring only one idea.

**Response Format:**
```
{
  "score": <1-5>,
  "reasoning": "<brief explanation of your score>",
  "categories_observed": ["<category 1>", "<category 2>", ...]
}
```
**Important:** Focus on semantic diversity, not visual or stylistic similarity. Consider whether the items explore different facets or interpretations of the original query.

---

where KL penalties are applied directly to the per-token loss. The policy is optimized to generate $k$ sub-queries that maximize task-specific set-level rewards, including diversity, groundedness, and alignment. We use a learning rate of $1 \times 10^{-7}$ and a global batch size of 512. Training was conducted on a TPUv6e-16 Ghostlite Pod (consisting of 16 accelerator cores).

**Supervision Synthesis.** Following RL optimization, we use the trained FOLM to synthesize objective-consistent training pairs. For each query, we generate 128 samples at a temperature of 0.9 to capture the reward-shaped distribution of fan-out behaviors. This data generation stage was performed on a TPUv6e-4 Ghostlite Pod (4 accelerator cores).

**Diffusion Training.** The diffusion model employs a Transformer backbone adapted for continuous inputs, where the input is a concatenated sequence of target embeddings $Z_{target} \in \mathbb{R}^{L \times d}$. The model is trained to predict $Z_0$ directly using a Variance Exploding (VE) formulation within the EDM framework. We utilize a Diffusion Transformer with a hidden dimension of 1024 and an MLP dimension of 1024, optimized with a warmup cosine decay sched-

ule over $10 \times 10^6$ steps. At inference, the model generates $L$ embeddings in a single pass using a SDE solver for 256 steps, which are then mapped to database contents via nearest-neighbor retrieval. This training was conducted on a TPUv6e-16 Ghostlite Pod (16 accelerator cores).

The specific parameters used for our experiments are detailed in Table 3.

---

**OAR - Query-Collection Alignment Evaluation**

**System Prompt:**
You are an expert evaluator assessing how well a set of retrieved [Image/audio] *collectively* aligns with an original query. User Prompt: You will be shown sub-queries and their representative [Image/audio]. Please evaluate how well this **entire set of [Image/audio]** represents the **original query**.

**User Prompt:**
`Original Query:` {query}
`Retrieved Items:` [Images/audio are presented in random order]

Please evaluate the **overall alignment** of this [Image/audio] set to the **Original Query** on a scale of 1-5:

**Evaluation Criteria:**

- **5 (Excellent):** Perfectly relevant. The set of [Image/audio] perfectly captures the full meaning and intent of the original query.

- **4 (Very Good):** Highly relevant. All [Image/audio] are clearly related to the query and capture its intent well.

- **3 (Good):** Relevant. The set of [Image/audio] clearly relates to the original query. Most images are on-topic. (This is the expected 'good' result).

- **2 (Fair):** Weakly relevant. A few [Image/audio] relate to the query, but the set as a whole is off-topic or misses the main idea.

- **1 (Poor):** Mostly irrelevant. The set of [Image/audio], as a whole, does not seem related to the original query.

**Response Format:**

```
{
  "score": <1-5>,
  "reasoning": "<brief explanation of your score>",
  "relevant_items_count": <number>,
  "total_items_count": <number>
}
```

**Important**: Judge the set as a whole. Do these [Image/audio] give you a good understanding of the original query?

## OAR - Groundedness Evaluation

**System Prompt:**
You are an expert evaluator assessing how well a retrieved [Image/audio] matches its **intermediate sub-query**.

**User Prompt:**
You will be shown sub-queries that were generated to retrieve items, along with the items actually retrieved for each sub-query.
`Sub-query 1:` {subquery_1} `Retrieved Item 1:` [Image/audio]
`Sub-query 2:` {subquery_2} `Retrieved Item 2:` [Image/audio]
... [repeat for all $k$ sub-queries]

For each sub-query and its retrieved item, evaluate the **groundedness** on a scale of 1-5:

**Evaluation Criteria:**

- **5 (Excellent):** Perfect match. The [Image/audio] is a perfect, textbook example of its sub-query.

- **4 (Very Good):** Strong match. The [Image/audio] clearly and specifically illustrates the sub-query's concept.

- **3 (Good):** Good match. The [Image/audio] is a clear and reasonable example of its sub-query. (This is the expected 'good' result).

- **2 (Fair):** Weakly related. The [Image/audio] is on the general topic (e.g., 'schoolcore') but fails to show the sub-query's specific concept (e.g., 'velvet').

- **1 (Poor):** Total mismatch. The [Image/audio] is completely unrelated to its sub-query.

**Response Format:**

```
{
  "per_subquery_scores": [
    {
      "subquery": "<subquery_text>",
      "score": <1-5>,
      "reasoning": "<brief explanation>"
    },
    ...
  ],
  "average_score": <1-5>
}
```

**Important:** You are NOT judging alignment to the original query. You are ONLY judging if the [Image/audio] for 'Sub-query 1' matches 'Sub-query 1'.

| Model | Sub Query | Retrieved Images |
|---|---|---|
| R4T | bohemian labor day picnic outfit |  |
| R4T | minimalist labor day picnic outfit |  |
| R4T | labor day picnic jumpsuit |  |
| Gemini 2.5 Flash | casual labor day picnic outfit |  |
| Gemini 2.5 Flash | stylish labor day picnic outfit |  |
| Gemini 2.5 Flash | comfortable labor day picnic outfit |  |

*Figure 6.* Examples of generated subqueries and retrieved images from Polyvore dataset given a broad query "Labor day picnic outfit" for the open-ended abstract retrieval task.

*Table 3.* Implementation Hyperparameters for R4T.

| Parameter | Value |
|---|---|
| ***Fan-Out LM (RL Training)*** | |
| Optimizer | AdamW ($\beta_1 = 0.9, \beta_2 = 0.95$) |
| Learning Rate | $1 \times 10^{-7}$ |
| Global Batch Size | 512 |
| Micro-batch Size | 64 |
| Grad. Accum. Steps | 8 |
| Max Seq. Length | 1024 |
| Group Size ($G$) | 8 |
| Clip Epsilon ($\epsilon$) | 0.2 |
| Forward KL Coeff. ($\beta_1$) | 0.05 |
| Reverse KL Coeff. ($\beta_2$) | 0.05 |
| Reward Norm. | Group-Standardized |
| Advantage Est. | Group-Relative |
| ***Diffusion Training*** | |
| Model Type | Coherent Transformer |
| Seq. Length ($L$) | 12 |
| Embed. Dim ($d$) | 128 |
| Hidden Dim | 1024 |
| MLP Dim | 1024 |
| Heads | 16 |
| Layers | 6 |
| Dropout / Attn. Dropout | 0.1 |
| Optimizer | Adam |
| Schedule | Warmup Cosine Decay |
| Peak LR | $3 \times 10^{-4}$ |
| Warmup Steps | 20,000 |
| Total Steps | $10 \times 10^{6}$ |
| Batch Size | 512 |
| EMA Decay | 0.9999 |
| Scheme | Variance Exploding |
| Weighting | EDM |
| Noise Schedule | Tangent |
| Range $[\sigma_{\min}, \sigma_{\max}]$ | $[10^{-4}, 80.0]$ |
| Data Std ($\sigma_{\text{data}}$) | 0.088 |
| Cond. Drop ($p_{\text{drop}}$) | 0.1 |
| CFG Strength | 0.1 |
| Steps | 256 |

