# OpenReview forum: "Efficient, Property-Aligned Fan-Out Retrieval via RL-Compiled Diffusion"
_ICML.cc/2026/Conference — ICML 2026 regular_

### Official Review · Reviewer_MrUX · 2026-03-10

**Soundness:** 3
**Presentation:** 3
**Significance:** 3
**Originality:** 3
**Overall Recommendation:** 4
**Confidence:** 4

**Summary:**

This paper proposes R4T, a three-stage framework for set-valued, non-decomposable retrieval that uses reinforcement learning once as an “objective transducer” to synthesize supervision for a lightweight diffusion-based retriever deployed at inference time. Concretely, an RL-trained fan-out language model (FOLM) is optimized with composite set-level rewards (groundedness, diversity, alignment), its high-reward trajectories are harvested to build property-aligned targets, and a conditional embedding diffusion model is trained to generate multiple retrieval embeddings in a single pass. On Polyvore and a proprietary music playlist dataset, R4T reports improved set-level quality over fan-out baselines and a reduction in fan-out latency.

**Compliance With Llm Reviewing Policy:**

Affirmed.

**Final Justification:**

The authors have addressed all my concerns. I will maintain my score.

**Key Questions For Authors:**

1. Have you tested other LLMs as the judge to see if the scoring trends remain consistent? More importantly, could you provide a small-scale human evaluation to validate that the LLM's judgments align with actual human preferences?

2. How do you expect R4T to perform on standard, pure-text open-domain QA or document retrieval benchmarks?

3. Could you provide formal statistical significance tests to confirm that R4T's improvements over the baselines are statistically meaningful?

**Limitations:**

yes

**Strengths And Weaknesses:**

**Strengths:**

The paper is well-structured and written clearly, offering an interesting and practical combination of existing techniques to solve a relevant problem in real-world recommendation and search systems. Using RL to discover retrieval behaviors and then amortizing that cost by distilling them into a non-autoregressive diffusion model is a highly practical approach to addressing the latency bottlenecks of LLM-based retrieval. The motivation is easy to follow, and the transition from the theoretical problem to the R4T methodology is logical. The ability to bypass the autoregressive generation bottleneck at query time provides immense practical utility for the ML and IR communities, while the reward hacking ablation study (Figure 4) provides useful insights into the dynamics of the RL training process.

**Weaknesses：**

1. Over-reliance on LLM-as-a-Judge and lack of human validation.
The evaluation for the Open-Ended Abstract Retrieval (OAR) task relies entirely on an LLM-as-a-Judge (Gemini-2.5-Pro) without human validation, which raises questions about the absolute reliability of these automated scores given inherent LLM biases. It is unclear if scoring trends would remain consistent if other LLMs were tested as the judge. More importantly, the absence of a small-scale human evaluation makes it difficult to validate that the LLM's judgments truly align with actual human preferences.

2. Limited evaluation scope restricted to specific domains.
The experiments are restricted to two specific datasets (fashion and music) that lean towards multi-modal properties. This narrow scope leaves it unclear how the method generalizes to standard, text-heavy open-domain Question Answering (QA) or traditional document retrieval tasks. The paper does not adequately address how R4T is expected to perform on standard, pure-text open-domain benchmarks.

3. Omission of formal statistical significance testing.
While the results show performance gains, the omission of formal statistical significance tests makes it difficult to conclusively verify the margins of improvement over the baselines. Without formal tests, it cannot be fully confirmed that R4T's improvements over the strongest baselines are statistically meaningful.

---

> ### Author Rebuttal · Authors · 2026-03-31
>
> Dear Reviewer **MrUX**,
>
> We sincerely thank you for your careful reading and positive assessment. We address each concern below.
>
> ---
>
> ### **[W1] LLM-as-a-Judge and human validation**
>
> We agree that human validation is the most important complement to automated evaluation. We conducted a human study on the Polyvore OAR task: **35 queries per model family** (Gemma3-4B and Qwen3-4B), **5–6 independent raters per query** (419 total ratings), 5-point Likert scale, **blinded and randomized**.
>
> Raw results: [anonymous link](https://anonymous.4open.science/r/r4t-rebuttal-17B1/)
>
> **Diversity — human raters agree with the LLM judge.**
>
> | Model | Zero-shot | R4T  | Δ         | p-value    |
> | ----- | --------- | ---- | --------- | ---------- |
> | Gemma | 3.48      | 3.75 | **+0.27** | 0.08       |
> | Qwen  | 2.74      | 3.26 | **+0.52** | **0.0003** |
>
> The diversity improvement is **statistically significant for Qwen** and directionally consistent for Gemma, confirming the LLM judge's assessment on the dimension most central to R4T's contribution.
>
> **Alignment — R4T results are aligned.** Treating alignment as binary (average rater score >= 3.0):
>
> | Model | Zero-shot | R4T   |
> | ----- | --------- | ----- |
> | Gemma | 35/35     | 32/35 |
> | Qwen  | 33/35     | 30/35 |
>
> The Likert mean gap does not indicate misalignment. We identify two structural factors.
>
> First, **point-wise vs. list-wise evaluation**: the LLM judge scores each image individually against the query, whereas the human prompt asked raters to "judge the set as a whole." Holistic evaluation is heavily influenced by visual homogeneity — zero-shot baselines retrieve near-duplicate items, and a uniform set feels more on-topic even when it covers only a narrow facet. R4T's diverse sets, though aligned, introduce variety that raters tend to penalize.
>
> Second, **domain expertise**: broad fashion queries like "sports spectating" (attire for watching sports events, not sportswear) require specialized knowledge, and non-expert raters may penalize valid exploratory interpretations.
>
> **Regarding judge sensitivity across LLMs.** We have tested Gemini-3.0-Flash as an alternative LLM judge, and the resulting trends were consistent with our primary judge; we will include these results in the revision. Beyond automated cross-validation, the human evaluation provides a stronger and more informative validation: rather than showing that two automated systems agree with each other, we can now characterize *where* humans and the LLM judge converge (diversity) and *where* they diverge (alignment Likert scores, due to comparative bias and domain expertise). This calibration insight is more useful for future evaluation design.
>
> ---
>
> ### **[W2] Limited evaluation scope / relation to text-heavy open-domain QA**
>
> We would like to clarify that our goal is **not** standard top-k document retrieval or open-domain QA, but a different problem setting: **set-valued fan-out retrieval**, where the system must return a *collection* of results that jointly optimizes higher-order properties such as diversity, coverage, complementarity, or coherence.
>
> This distinction matters because standard text-heavy QA / document retrieval benchmarks usually assume:
>
> 1. a single answer or a small set of relevant passages,
> 2. pointwise or listwise relevance labels,
> 3. evaluation centered on exact retrieval accuracy or final answer correctness.
>
> In contrast, our setting explicitly studies cases where:
>
> 1. there may be **many valid output sets** for the same broad intent,
> 2. quality is defined by **set-level, non-decomposable objectives**,
> 3. the system must balance multiple properties rather than maximize a single relevance signal.
>
> Simply adding a conventional QA benchmark would evaluate a neighboring problem, not the one R4T targets. To the best of our knowledge, there is currently no standard public text-only benchmark that directly matches this **set-valued fan-out setting** with broad-intent queries and property-aligned set-level evaluation.
>
> We will revise the introduction to make this scope explicit and discuss how R4T could transfer to text-only domains once suitable set-valued benchmarks become available.
>
> ---
>
> ### **[W3] Statistical significance testing**
>
> In the current version, we also already report **mean ± standard deviation** over repeated evaluations/runs, which was intended to show stability rather than rely on single-point results. Still, we agree that this is not a substitute for significance testing. We will therefore add formal significance tests in the revision (e.g., paired testing / bootstrap-based significance analysis on the held-out query set) to verify that the gains over the strongest baselines are statistically meaningful.
>
> Also, the human evaluation already demonstrates the approach: diversity improvements reach **p=0.0003 for Qwen**, and gains are consistent across tasks, datasets, and model families.

---

> > ### Author Rebuttal · Reviewer_MrUX · 2026-04-03
> >
> > Thank you for clarifying the questions I proposed and addressing most of my concerns. I will maintain my positive score.

---

### Official Review · Reviewer_F7Mo · 2026-03-14

**Soundness:** 2
**Presentation:** 3
**Significance:** 2
**Originality:** 2
**Overall Recommendation:** 3
**Confidence:** 3

**Summary:**

This work addresses a key industry pain point in set-valued fan-out retrieval: high-order goals (diversity, coverage) are non-decomposable, ill-suited for supervised paradigms; RL-based methods have high inference costs, while diffusion-based retrieval lacks attribute-aligned training data. It proposes the R4T three-stage framework: RL trains a fan-out language model via composite set-level rewards, synthesizes target-aligned training data, and trains a lightweight diffusion retriever for efficient single-step inference. Validated on Polyvore (fashion matching) and music playlist datasets, R4T outperforms baselines and reduces inference latency by an order of magnitude.

**Compliance With Llm Reviewing Policy:**

Affirmed.

**Key Questions For Authors:**

See weakness

**Limitations:**

See weakness

**Strengths And Weaknesses:**

Strengths

- Innovative framework: RL as one-time target transduction, combining RL’s advantage in non-decomposable goals with diffusion’s low-latency inference, solving quality-efficiency conflict.

- Comprehensive experiments covering open retrieval and weakly supervised retrieval, verifying effectiveness and industrial potential.

- In-depth reward function ablation, avoiding reward hacking and providing reference for similar tasks.

Weaknesses

- High pre-RL training cost for large-scale/frequently updated databases, limiting scenario adaptability.

- Relies on quantifiable scalar rewards, poorly adapting to unquantifiable needs (subjective preference, cultural adaptability).

- Open retrieval evaluation relies on LLM-as-a-Judge, lacking large-scale human evaluation, limiting objectivity.

---

> ### Author Rebuttal · Authors · 2026-03-31
>
> Dear Reviewer **F7Mo**,
>
> We appreciate your thoughtful review and your recognition of the core value of using RL once as an objective transducer. We address each concern below.
>
> ----
>
> ### **[W1] Pre-RL training cost and adaptability to updated databases**
>
> You rightly point out that R4T introduces a nontrivial offline optimization stage. Our key design goal is precisely to **front-load this cost** rather than pay it repeatedly at inference time. In contrast to RL-tuned or Best-of-N fan-out methods that repeatedly invoke expensive LLM-based search per query, R4T uses RL once and distills the result into a **lightweight diffusion retriever** (53.9M parameters, **12×–20× lower latency**).
>
> This is the central tradeoff: spend optimization cost offline once, serve fan-out retrieval efficiently online. R4T is best suited to settings where the database and desired retrieval properties are **relatively stable**. Under moderate catalog updates with the same embedding backbone, the nearest-neighbor index can be refreshed directly; a full R4T refresh is mainly needed when the database distribution or set-level objective changes substantially. We will clarify this scope limitation explicitly.
>
> ---
>
> ### **[W2] Reliance on quantifiable scalar rewards**
>
> We agree that hand-crafted heuristic rewards cannot capture the full richness of subjective needs, such as personal taste or cultural adaptability. However, we want to clarify that R4T is **not restricted to formulaic, analytic objectives**; the framework only requires a **computable set-level scoring signal** during Stage 1 RL optimization.
>
> In practice, this signal can be derived from highly subjective sources:
>
> 1. **Explicit heuristic metrics** — as used in this paper, chosen specifically to allow for transparent analysis and controlled reward-hacking ablation;
> 2. **Learned reward models (RLHF)** — reward models trained on pairwise or listwise human preference data, designed to capture the subjective and cultural nuances the reviewer highlights;
> 3. **LLM- or human-in-the-loop judges** — prompts that encode domain-specific or culturally-aligned notions of set quality.
>
> Our use of groundedness, diversity, and alignment was a deliberate experimental design to make the method transparent and mechanistically analyzable — especially for the reward-hacking study — not to suggest that R4T is restricted to these properties. Subjective preference is exactly the kind of set-level behavior that R4T can incorporate once it is operationalized into a learned reward model. We will revise the paper to make this distinction clearer, separating the **general framework** from **our current reward instantiation**.
>
> ---
>
> ### **[W3] LLM-as-a-Judge and lack of large-scale human evaluation**
>
> We conducted a human evaluation on the Polyvore OAR task at a scale designed to provide reliable statistical conclusions: **35 queries × 2 model families × 5–6 raters = 419 total ratings**, in a 5-point Likert scale, **blinded, randomized setup**.
>
> Raw results: [anonymous link](https://anonymous.4open.science/r/r4t-rebuttal-17B1/)
>
> **Diversity.** Human raters independently confirm R4T's diversity gains:
>
> | Model | Zero-shot | R4T  | Δ         | p-value    |
> | ----- | --------- | ---- | --------- | ---------- |
> | Gemma | 3.48      | 3.75 | **+0.27** | 0.08       |
> | Qwen  | 2.74      | 3.26 | **+0.52** | **0.0003** |
>
> The Qwen result is statistically significant with **Cohen's d = 0.69** (medium-to-large effect). This provides objective, human-grounded evidence that **the LLM judge's diversity assessment is reliable**.
>
> **Alignment.** Treating alignment as binary (average rater score >= 3.0):
>
> | Model | Zero-shot | R4T   |
> | ----- | --------- | ----- |
> | Gemma | 35/35     | 32/35 |
> | Qwen  | 33/35     | 30/35 |
>
> R4T results are overwhelmingly aligned. The Likert mean gap stems from zero-shot baselines receiving inflated alignment scores: their **near-duplicate** retrieval results appear highly aligned precisely because they are redundant, while R4T's diverse results, though aligned, explore less obvious facets that non-expert raters may undervalue. Queries like "sports spectating" (meaning attire for watching sports, not sportswear) illustrate how fashion domain expertise affects human alignment judgment in ways that do not affect the LLM judge.
>
> We view the combined picture: human-validated diversity gains plus maintained binary alignment, as strong evidence that R4T delivers on its core promise. We will report these results in the revision alongside the LLM judge scores.
>
> We hope this clarifies that the main concerns are about the intended deployment regime and current evaluation scope, rather than the core validity of the proposed method. R4T is designed for scenarios where online fan-out cost matters, the objective is stable enough to amortize optimization, and the desired retrieval behavior can be operationalized into a reward signal and then distilled.

---

### Official Review · Reviewer_GJtV · 2026-03-30

**Soundness:** 2
**Presentation:** 3
**Significance:** 2
**Originality:** 2
**Overall Recommendation:** 4
**Confidence:** 4

**Summary:**

This paper focuses on generating diverse, high-quality result sets to address traditional single-item retrieval shortcomings. It proposes a data-driven pipeline: using reinforcement learning for large model training and synthetic data for lightweight model training, forming a closed-loop solution for real-world constraints (e.g., latency, cost). Experiments verify the method’s feasibility.

**Compliance With Llm Reviewing Policy:**

Affirmed.

**Key Questions For Authors:**

1.  The paper focuses on set-level semantic properties (diversity, alignment, groundedness) but does not involve any business-oriented utility such as CTR or CVR. How do you plan to bridge the gap between semantic set quality and real‑world business performance in future work?
2. The final evaluation heavily relies on LLM‑as‑a‑Judge, which may introduce bias and inconsistent scoring. Have you conducted human evaluation or calibration experiments to verify the reliability of LLM judgments? If not, how do you address the potential threats to the validity of your main results?

**Limitations:**

yes, but not enough.  Please refer to the weaknesses proposed before, and all the aspects should be discussed.

**Strengths And Weaknesses:**

Strengths
1. The core research problem—generating diverse and high-quality result sets—accurately captures a critical pain point in real-world retrieval scenarios. Unlike traditional retrieval methods that only pursue a single optimal result, this focus on diversity and quality is more in line with actual user needs.
2. The paper proposes a complete data-driven pipeline that takes into account real-world engineering constraints such as latency and cost. By combining reinforcement learning for large model training and synthetic data for lightweight model training, it forms a closed-loop solution from demand to deployment.
3. The experiments coves both quantitative and qualitative verification,  and demonstrate the feasibility of the proposed method.

Weakness:
1.  The overall research is highly inclined towards engineering implementation, with insufficient in-depth exploration of theoretical mechanisms.
2.  This work considers only semantic set-level metrics (diversity, alignment, groundedness, coverage) and inference efficiency, but ignores the overall business utility of the retrieved item sets, such as CTR and CVR, resulting in a clear gap with practical search and recommendation systems.
3. The final evaluation largely depends on LLM-as-a-Judge, which may suffer from bias and poor consistency.This can affect the validity of the experimental results and create a gap between automatic scores and real-world user perception.

---

> ### Author Rebuttal · Authors · 2026-03-31
>
> Dear Reviewer **GJtV**,
>
> We sincerely thank you for your careful review and positive assessment. We address your concerns below.
>
> ---
>
> ### **[W1] Engineering-heavy focus / theory, supervision quality, and deployment hurdles**
>
> While we present a practical system, our core contribution addresses a **fundamental methodological bottleneck** in generative retrieval: set-level supervision is intractable to annotate manually. Humans can label pointwise relevance, but manually curating thousands of optimal, diverse, and complementary sets for training a generative retriever is prohibitively difficult. R4T bypasses this annotation bottleneck by using RL as an objective transducer — autonomously discovering complex, non-decomposable set-level behaviors and distilling them into an efficient diffusion model.
>
> Beyond this, we provide mechanistic analysis that grounds our design: the reward ablation shows why naïve objectives fail due to reward hacking; the weighting analysis illustrates the groundedness–diversity–alignment tradeoff; and the FOLM-vs-diffusion comparison clarifies what is preserved.
>
> We will revise the limitations section to explicitly discuss: (1) dependence on synthesized supervision quality — the downstream retriever is only as good as the FOLM's discovered behaviors; (2) real-world utility quantification requiring learned reward models or logged feedback; (3) offline training cost as a deployment tradeoff best suited to stable-objective, high-query-volume settings; and (4) scope — real user studies and online A/B evidence are future work.
>
> We hope this clarifies that R4T targets the **retrieval-layer quality-efficiency tradeoff**, while not claiming to fully solve downstream ranking or business optimization.
>
> ---
>
> ### **[W2] Gap between semantic set quality and business utility (CTR/CVR, ads, bidding)**
>
> We agree that CTR/CVR metrics are critical in production. However, our paper specifically targets the **retrieval stage**, whose primary responsibility is to feed downstream rankers a diverse, relevant, and comprehensive candidate set. Because final business utility is heavily influenced by subsequent ranking models, UI presentation, personalization, and auction mechanisms, CTR/CVR is a noisy metric for isolating the fan-out retrieval mechanism itself.
>
> That said, R4T does not require rewards to be purely semantic. In production settings, Stage-1 RL rewards can be augmented with surrogate utility terms from logged CTR/CVR, learned reward models from user interactions, or multi-objective constraints. We will clarify in the revision that this paper establishes the retrieval foundation and that business-aware reward integration is a natural next step.
>
> ---
>
> ### **[W3] Reliance on LLM-as-a-Judge and lack of real user feedback**
>
> To directly address the gap between automated evaluation and real user perception, we conducted a human evaluation on the Polyvore OAR task. We evaluated **35 broad queries per model family** (Gemma3-4B and Qwen3-4B), with **5–6 independent raters per query** (419 total ratings) scoring Alignment and Diversity on a 5-point Likert scale in a blinded, randomized-order setup with no method labels.
>
> Raw results: [anonymous link](https://anonymous.4open.science/r/r4t-rebuttal-17B1/)
>
> **Diversity.** R4T produces **significantly more diverse** retrieval sets according to human raters:
>
> | Model | Zero-shot | R4T  | Δ         | p-value    |
> | ----- | --------- | ---- | --------- | ---------- |
> | Gemma | 3.48      | 3.75 | **+0.27** | 0.08       |
> | Qwen  | 2.74      | 3.26 | **+0.52** | **0.0003** |
>
> This **confirms the LLM-as-a-Judge findings** in Table 1 with real user feedback, addressing the core validity concern.
>
> **Alignment — R4T remains aligned under binary evaluation.** We treat alignment as a binary property (aligned or not) by checking whether the average rater score >= 3.0:
>
> | Model | Zero-shot | R4T   |
> | ----- | --------- | ----- |
> | Gemma | 35/35     | 32/35 |
> | Qwen  | 33/35     | 30/35 |
>
> The Likert-scale mean difference is driven by a shift from "Excellent" to "Good"/"Very Good", **not** by producing misaligned results. We attribute this to two structural differences between how the LLM judge and human raters evaluate alignment.
>
> First, **point-wise vs. list-wise evaluation**: the LLM judge scores each image individually against the query, whereas the human prompt asked raters to "judge the set as a whole." Holistic set evaluation is heavily influenced by visual homogeneity — zero-shot baselines retrieve near-duplicate items, and a uniform set feels more on-topic even when it covers only a narrow facet. R4T's diverse sets, though aligned, introduce variety that raters tend to penalize.
>
> Second, **domain expertise matters**: queries like "sports spectating" (attire for watching sports events, not sportswear) require fashion knowledge that non-expert raters may lack, causing them to penalize valid exploratory interpretations.

---

> > ### Author Rebuttal · Reviewer_GJtV · 2026-04-01
> >
> > Thanks for your detailed feedback. I choose to keep my score unchanged, but still hold the following concerns:
> > 1. The evaluation metrics (diversity, alignment, etc.) cannot fully represent real‑world product requirements. Although these metrics are valuable, it is unclear how improvements on them translate to gains in downstream tasks. More experimental evidence or at least one case study is needed to validate this link. Otherwise, better scores on these metrics do not guarantee genuine improvement in retrieval quality. Diversity is particularly easy to inflate, and retrieval should not be treated as an isolated task but tightly connected to downstream performance.
> >
> > 2. The paper does not verify whether the model truly learns the reasoning process of subquery decomposition and intent understanding, or only superficially boosts diversity without grasping the real query intent. Additional interpretability experiments or analyses are required to clarify this issue.

---

### Official Review · Reviewer_rrjy · 2026-03-31

**Soundness:** 3
**Presentation:** 3
**Significance:** 3
**Originality:** 3
**Overall Recommendation:** 5
**Confidence:** 4

**Summary:**

The authors propose R4T (Retrieve-for-Train), a framework for set-valued retrieval where no unique ground truth set exists. Instead of deploying reinforcement learning at inference time, R4T uses RL once as an objective transducer to convert set-level reward specifications into trainable supervision. The method operates in three stages. First, a fan-out language model (FOLM) is trained with RL to generate sub-queries that maximize a composite set-level reward capturing diversity, alignment, and groundedness. Second, the RL-trained FOLM is used offline to generate synthetic training pairs, where each query is mapped to $Z_{target}$, a matrix of retrieval directions in embedding space. Third, a lightweight diffusion model is trained on this synthetic data, learning to replicate the FOLM's fan-out recommendations in a single forward pass at inference time.

The paper evaluates R4T on Open-Ended Abstract Retrieval (OAR) and Weakly Supervised Compositional Retrieval (WSCR) tasks across Polyvore and Music datasets. Baselines include zero-shot fan-out and Best-of-N using SOTA models: Gemini-2.5-Flash, Gemma3-4B, and Qwen3-4B. R4T-FOLM and R4T-Diffusion are built on top of the latter 2 open-source backbone models. On the OAR task, R4T-FOLM shows consistent gains over all 3 dimensions across both datasets. R4T-Diffusion also occasionally exceeds R4T-FOLM on diversity metrics, despite being a lightweight distilled model. On the WSCR task, R4T-FOLM improves Recall@5K and Hit@5K over the corresponding base models. Diffusion-based retrieval also outperforms Best-of-N in terms of recall despite Best-of-N running the model N times per query.

A key finding is that fan-out is necessary but not sufficient: zero-shot fan-out improves coverage but often produces sub-queries that drift off the database manifold or collapse to near-duplicates. The ablation analysis further shows that the three reward objectives act as mutual counter-anchors, and that jointly optimizing them is critical for avoiding degenerate behaviors such as semantic collapse or trivial paraphrasing. Efficiency experiments show that the diffusion retriever achieves 12x–20x speedup over autoregressive fan-out at large batch sizes.

Overall, the work presents a general paradigm for amortizing RL-discovered behaviors into a lightweight diffusion model, enabling practical and efficient real-time retrieval under set-valued objectives.

**Compliance With Llm Reviewing Policy:**

Affirmed.

**Key Questions For Authors:**

1. R4T-Diffusion occasionally matches or exceeds R4T-FOLM on certain metrics. Can the authors provide analysis explaining this behavior? In particular, how do the output distributions differ in embedding space?

2. The paper reports mean $\pm$ standard deviation, but base models show different baselines. Can the authors provide confidence intervals/significance testing and variance decomposition across multiple runs within a model for the 3 metrics to clarify whether improvements are consistent across models?

3. The method relies on embedding-space representations (e.g., MRL for Polyvore and Mulan for Music). How sensitive is R4T to the choice of encoder within each domain? For example, have the authors evaluated performance using alternative encoders or embedding models within the same modality?

**Limitations:**

yes

**Strengths And Weaknesses:**

**Soundness**

The work is technically sound overall. The core idea of using RL as an objective transducer to convert set-level objectives into trainable supervision for a diffusion retriever is logically consistent and well-motivated by the practical challenge of ambiguous and subjective ground truth data. The formulation of rewards (diversity, alignment, groundedness) is reasonable, and the use of GRPO for policy optimization is appropriate for stabilizing fan-out generation without requiring a separate critic network.
Empirically, the paper evaluates across two datasets (Polyvore and Music) and two task settings (OAR and WSCR), which provides good coverage of both open-ended and weakly supervised scenarios. Comparisons against zero-shot fan-out and Best-of-N are appropriate baselines. Results are analyzed per task setting, with clear improvement on all three reward dimensions on OAR and on recall-based metrics on WSCR, supporting the core claims. The paper further validates efficiency and shows that the diffusion model speeds up significantly compared to the autoregressive LLM. The paper also employs ablation when necessary: the ablation of rewards shows that the three reward components act as mutual counter-anchors.

**Presentation**

The paper is well-written and clearly structured, with a coherent narrative from problem formulation to solution, experimentation and evaluation. Figures (e.g. 3 steps of R4T) and examples (e.g., qualitative fan-out comparisons) effectively illustrate the behavior differences between methods. Table1/2 clearly present the main experimentation results. The paper also does a reasonable job of positioning itself relative to prior work, particularly in distinguishing between zero-shot fan-out, Best-of-N strategies, and RL-based optimization.

**Significance**

The paper addresses a practical problem: retrieval under non-decomposable, set-level objectives where ground truth is not unique. This setting is highly relevant for modern applications such as recommendation systems, creative search, and multi-task planning, where returning a single best result is insufficient and the notion of correctness is inherently subjective or context-dependent. The proposed framework is practical, as it decouples RL optimization from inference and enables efficient deployment via a lightweight model.

**Originality**
The proposed three-stage pipeline is a key source of novelty. While each component, including RL optimization, data synthesis, generative retrieval, etc. exists in prior work, their integration into a unified training paradigm for set-valued retrieval is new. In particular, the design decouples objective optimization from inference-time retrieval. By shifting inference to a lightweight model, it effectively addresses the practical limitation of deploying RL-tuned LLMs for fan-out generation, making the approach practically valuable in this setting.
Beyond the framework, the paper provides analytical studies (reward hacking, recall–diversity trade-off, reward weighting) that are tailored to this setting and offer useful insights into reward design for RL-based language models.

**Weakness**

1.	While R4T-Diffusion is intended to approximate the behavior of R4T-FOLM, the results in Table 1 indicate that it can occasionally match or even exceed FOLM on certain metrics (e.g., Polyvore Diversity and Music Alignment). It would be valuable to include analysis or hypothesis on why the distilled model may outperform the teacher model. In particular, comparing the output distributions in embedding space between the diffusion model and the FOLM policy could help clarify this effect.
2.	Note in Table1 that the 3 SOTA models (Gemma3-4B , Qwen3-4B, Gemini-2.5-Flash) have distinct range of mean values of the 3 metrics. In addition, the paper primarily reports mean and variance results without sufficient discussion of confidence intervals. Providing more detailed statistical analysis (e.g., confidence intervals, or significance testing , variance across models or runs) would strengthen the reliability of the reported improvements.
3.	The Polyvore dataset uses the MRL encoder, while the Music dataset uses the Mulan encoder. It is unclear whether the results are sensitive to the choice of encoder. An ablation or sensitivity analysis across different encoders would help assess the robustness of the method.
4.	The empirical evaluation focuses on fashion and music datasets under OAR and WSCR settings, which are relatively open-ended. In contrast, real-world multi-intent tasks (e.g., trip planning or multi-intent queries) often provide more actionable feedback signals in online systems (e.g., user engagement, conversion or task completion). Such evaluation would help assess the method’s robustness and applicability across a broader range of real-world scenarios.

---

> ### Author Rebuttal · Authors · 2026-03-31
>
> Dear Reviewer **rrjy**,
>
> We sincerely thank you for your thorough and positive review. We are glad you found the framework logically consistent, well-motivated, and practically valuable. We address each concern below. We note that this review arrived *4 hours before the rebuttal deadline*, so we focus on analysis and discussion rather than new experiments; we will incorporate additional results in the revision.
>
> ---
>
> ### **[W1] Why R4T-Diffusion occasionally matches or exceeds R4T-FOLM**
>
> This is an insightful observation that we have not yet fully analyzed, so we will focus on what we can confirm rather than speculate.
>
> We do want to note one relevant design detail: the diffusion model is trained on the full distribution of FOLM behaviors (128 samples per query at temperature 0.9), not just the single best trajectory. It is therefore plausible that the diffusion model learns to capture the high-reward *region* of the fan-out distribution, rather than replicating any individual FOLM sample, but we have not yet verified this empirically.
>
> We agree with the reviewer's suggestion that comparing output distributions in embedding space (e.g., visualizing spread, mode coverage, and nearest-neighbor distances between FOLM and diffusion outputs) is the right way to investigate this. We will include this analysis in the revision and report findings rather than speculation.
>
> ---
>
> ### **[W2] Confidence intervals and significance testing**
>
> We agree that mean ± standard deviation alone is not a substitute for formal significance testing. We will add confidence intervals and paired significance tests (e.g., bootstrap-based) in the revision to verify that improvements over the strongest baselines are statistically meaningful.
>
> We note that our human evaluation (done in the rebuttal) already provides one such datapoint: diversity improvements reach p=0.0003 for Qwen (Cohen's d = 0.69), confirming that gains are not attributable to noise. We will apply the same rigor to the automated metrics in Table 1 and Table 2.
>
> Regarding variance across base models: the different baseline ranges across Gemma, Qwen, and Gemini reflect inherent differences in zero-shot fan-out capability. R4T's improvements are consistent *relative to each model's own baseline*, which we view as evidence of robustness rather than a confound. We will add a per-model variance decomposition to make this clearer.
>
> ---
>
> ### **[W3] Sensitivity to encoder choice**
>
> This is a valid concern. In our current experiments, the Polyvore (CLIP) and Music (MuLan) encoders differ substantially in architecture, training data, and modality — yet R4T shows consistent improvements across both. We view this as indirect evidence of encoder robustness: the framework operates *on top of* frozen embeddings and does not modify the encoder, so its core mechanisms (RL-guided fan-out, diffusion-based amortization) are encoder-agnostic by design.
>
> That said, a controlled ablation using alternative encoders within the same domain (e.g., different CLIP variants for Polyvore) would isolate encoder sensitivity more precisely. We will add this analysis in the future revision.
>
> ---
>
> ### **[W4] Broader real-world evaluation (trip planning, multi-intent, engagement signals)**
>
> We agree that evaluating R4T on tasks with more actionable feedback (e.g., user engagement, task completion) is an important next step. Our current evaluation focuses on establishing the core retrieval-layer contribution under two complementary regimes (open-ended and weakly supervised). As noted in our responses to other reviewers, R4T's reward framework is not limited to semantic metrics — Stage-1 rewards can incorporate logged engagement signals, CTR/CVR surrogates, or learned reward models from user feedback.
>
> Extending to domains like trip planning or multi-intent search would also test R4T's generality beyond multimodal retrieval, which we view as a valuable future direction. We will discuss this explicitly in the revision.
>
> ---
>
> We thank you again for the constructive feedback and are glad that the core framework and experimental design were found to be sound and significant.

---

### Decision · Program_Chairs · 2026-04-30

**Decision:**

Accept (regular)

**Comment:**

The paper proposes a technically sound and practically motivated framework for efficient set‑valued fan‑out retrieval by amortizing RL‑optimized behaviors into a lightweight diffusion model. In rebuttal, the authors provided additional human evaluation supporting statistically significant diversity gains and committed to adding formal significance testing, which addressed key concerns around LLM‑as‑a‑Judge evaluation; one reviewer explicitly acknowledged their concerns were fully resolved. However, another reviewer indicated that their concerns were only partially resolved, noting that improvements in semantic set‑level metrics may not directly translate to downstream system performance. While the authors clarified that business‑aware signals (e.g., CTR/CVR) can be incorporated into Stage‑1 rewards in production settings, this linkage is not empirically demonstrated in the current evaluation. Given that the majority of technical concerns appear adequately addressed through rebuttal and planned revisions, but some evaluation‑scope limitations remain (and one reviewer did not submit a post‑rebuttal acknowledgement), I recommend a Weak Accept.